# A Review on Human Comfort Factors, Measurements, and Improvements in Human–Robot Collaboration

**DOI:** 10.3390/s22197431

**Published:** 2022-09-30

**Authors:** Yuchen Yan, Yunyi Jia

**Affiliations:** Department of Automotive Engineering, International Center for Automotive Research at Clemson University, Greenville, SC 29607, USA

**Keywords:** human comfort, human–robot collaboration, comfort measurement, comfort improvement, wearable sensing

## Abstract

As the development of robotics technologies for collaborative robots (COBOTs), the applications of human–robot collaboration (HRC) have been growing in the past decade. Despite the tremendous efforts from both academia and industry, the overall usage and acceptance of COBOTs are still not so high as expected. One of the major affecting factors is the comfort of humans in HRC, which is usually less emphasized in COBOT development; however, it is critical to the user acceptance during HRC. Therefore, this paper gives a review of human comfort in HRC including the influential factors of human comfort, measurement of human comfort in terms of subjective and objective manners, and human comfort improvement approaches in the context of HRC. Discussions on each topic are also conducted based on the review and analysis.

## 1. Introduction

In modern manufacturing, robots have become a critical and irreplaceable role which greatly reduces human physical labor as well as lowers the cost of factory operation. However, even though the automation industry technology has made enormous breakthroughs in the past few decades, most of today’s industrial robots are still used inside heavy fence guarding and safety peripheral equipment that is costly, inflexible, and bulky; normally, they have configured fixed infrastructure and have to use extra floor space [1]. Light curtains are commonly used for emergency protection; the entire system will shut off immediately as the human worker gets into the working area. Undoubtedly, a protection system like such is clumsy and inefficient in guaranteeing the safety of the human worker.

The Human–Robot Collaboration (HRC), known as “the state of a purposely designed robotic system and operator working in a collaborative workspace” [2], has gained growing attention in its research field during the past few years. It is an interdisciplinary field that focuses on the collaboration of humans and robots as they achieve shared goals [3].

Collaborative robots, also known as COBOTs, provide prospective and great solutions to complex hybrid assembly tasks, especially in smart manufacturing contexts [4]. Based on the concept of HRC, robot manufacturers have released different collaborative robots into the market, including some popular models like ABB Yumi, UR3, Kuka IIWA, etc. Through human–robot interaction, the tasks can be split between humans and robots based on their capabilities to leverage their unique advantages [5,6].

Even though these COBOTs have been proved to be highly capable and efficient in human–robot collaboration scenarios in many real-world applications, their usage and acceptance in the real world still remain very limited and have huge space for further improvement. The promotion of collaborative robots does not depend merely on the efficiency, flexibility, and intelligence of the robots. User acceptance is also an important factor. Currently, the user acceptance of collaborative robots is still low due to a list of concerns from different perspectives [7,8]. Among the factors which may affect user acceptance such as safety, trust and robot performance, human comfort is usually less emphasized in COBOT development but however critical to the user acceptance during HRC.

Better comfort perceived by the user can benefit the overall user acceptance of collaborative robots from a user perspective [9]. The comfort of humans plays such a critical role that it not only affects user acceptance but also has a significant impact on the efficiency of manufacturing [10,11,12]. Therefore, this paper gives a review of human comfort in HRC including the influential factors of human comfort, measurement of human comfort, and improvement approaches of human comfort.

Human comfort has been studied for decades from different fields, primarily in psychology. Webster Dictionary (1981) gave the definition of comfort as a “state or feeling of having relief, encouragement, and enjoyment”. Slater [13] proposed a more scientific definition of comfort in his book “Human Comfort”, which also includes the influence of environment. Slater defined comfort as a pleasant state of physiological, psychological, and physical harmony between a human being and its environment. Some researchers perceived comfort as two discrete states: comfort presence and comfort absence, where comfort is simply considered as the complete opposite side of discomfort, or in other words, the absence of discomfort. However, some other researchers held a contrasting opinion against the discrete state theory. They claim that comfort and discomfort are two opposites on a continuous scale, ranging from extreme discomfort through a neutral state to extreme comfort [14,15]. There are also some researchers disagreeing with this single-dimension continuous scale idea of comfort definition. Kamijo et al. [16] claimed that comfort and discomfort are affected by distinctly different variables. There are also some researchers who view comfort as an optimal state in which the person stops taking action to avoid discomfort [17].

Despite all the arguments and debates in the field, some consensus has been achieved on several points of view: (1) comfort is subjectively determined by each individual’s personal nature; (2) comfort can be affected by a wide variety of factors from multiple natures such as physical, physiological or psychological; and (3) comfort is affected by one’s reaction to the environment stimulus. This review paper focuses on human comfort defined as a feeling of ease in human–robot collaboration contexts. It can been seen that comfort is really a complex topic. A review to understand human comfort in the context of HRC is going to be very helpful to facilitate the research and development of HRC and their applications in the real world.

The rest of this review paper is structured into the following sections: Section 2 will introduce the influential factors on comfort in human–robot interaction; Section 3 will introduce the measurement approaches of human comfort including both subjective and objective approaches; Section 4 will introduce the improvement approaches of human comfort; and Section 5 will give the conclusions.

## 2. Review on Influential Comfort Factors

To better understand and study human comfort, it is crucial to know what factors could have impacts on comfort and how these factors affect human subjective feelings both qualitatively and quantitatively. Generally, the human comfort factors under HRC scenarios can be divided into several categories such as ergonomic factors, robot motion-based factors, anthropomorphism and robot sociability factors, etc.

### 2.1. Ergonomic Factors

Ergonomic factors studied in traditional factory working scenarios will maintain their influences in human–robot collaboration tasks. In a factory working environment, specifically speaking, under HRC scenarios, the two most critical ergonomic factors are noise and thermal.

#### 2.1.1. Noise

In a typical manufacturing plant packed with operating machines on the production lines, noise is one of the major health risks and also a major influential factor on workers’ comfort. Noises not only induce unpleasant feelings in humans but can also cause a variety of health issues, both auditorily and non-auditorily. Constantly being exposed to noise of above 85 dBA can lead to acoustic trauma, tinnitus, hearing loss, cardiovascular disease, etc. [18,19].

The two most critical parameters contributing to noise occupational health issues are the sound pressure level and the exposure time [18]. Either extremely high-pressure level with short duration noise, or relatively high-pressure level but long-term exposure noises could cause health issues. In order to eliminate or reduce the noises to improve human comfort, the main noise sources need to be localized, and the noise propagation needs to be analyzed before taking any actions to decrease exposure time and sound pressure level. This is the process of establishing a complete noise model, while all the interactions such as sound reflection and absorption between noise sources and the environment should be considered. Guarnaccia et al. [18] used the acoustic predictive software RAP-ONE to establish the noise map of the indoor testing environment. In Ouis’s review paper [20], annoyance is considered the major discomfort component induced by noises. Many research efforts have been put into building models to predict how annoyance varies with respect to noise exposure. Hall et al. [21] created a model demonstrating how activity interference affects the probability of annoyance. Izumi and Yano [22] developed a ‘path analysis’ to explain the annoyance responses obtained from questionnaires. The U.S. Air Force conducted a study on the relationship between the percentage of the population that is highly annoyed (% HA) and the day-night average sound level (DNL), and eventually formulated an equation that quantitatively describes this relationship. Pennig et al. [23] found human subjective pleasant level decreases in a linear pattern with respect to noise pressure level in an aircraft cabin environment. To improve acoustic comfort, annoying noises should be eliminated.

#### 2.1.2. Thermal

In order to quantitatively evaluate thermal comfort and analyze the heat transfer process between the human body and the surrounding environment, the definition of thermal comfort is given by ASHRAE Standard 55 as “the condition of mind that expresses satisfaction with the thermal environment and is assessed by subjective evaluation” [24]. Thermal comfort is a subjective attribute based on the net heat transfer between the human body and the environment. To quantitatively describe this thermal model, the ASHRAE Handbook of Fundamentals has also proposed an energy balance equation for the human body [25].
(1)$$M-W=\left(C+R+E_{sk}\right)+\left(C_{res}+E_{res}\right)+\left(S_{sk}+S_{cr}\right),$$
where

M= rate of metabolic heat production, W/m${}^{2}$;W= rate of mechanical work accomplished, W/m${}^{2}$;C+R= sensible heat loss from skin, W/m${}^{2}$;$E_{sk}=$ total rate of evaporative heat loss from skin, W/m${}^{2}$;$C_{res}=$ rate of convective heat loss from respiration, W/m${}^{2}$;$E_{res}=$ rate of evaporative heat loss from respiration, W/m${}^{2}$;$S_{sk}=$ rate of heat storage in skin compartment, W/m${}^{2}$;$S_{cr}=$ rate of heat storage in core compartment, W/m${}^{2}$.

This energy balance between the human body and the environment affects human subjective feelings of thermal comfort [26].

In terms of comfort evaluation methods, Ormuz et al. [27] developed a thermal mannequin equipped with a great number of sensors along the entire body, specifically used to assess the physical properties necessary to calculate the variables in the energy balance Equation (Equation 1).

Tiller et al. [28] studied the combined effects of noise and temperature on human comfort and performance by collecting subjective ratings from test subjects under multiple acoustical and temperature conditions and implementing statistical analyses. Tiller et al. found that the most preferred temperature range is within 72–76 °F, while other temperature conditions can create discomfort feelings. Huda [29] studied how thermal conditions’ change of the working environment affect the factory workers’ comfort and productivity. Huda calculated the Heat Stress Index (HSI) around the working area and analyzed worker productivity before and after the cooling system was installed, revealing that HSI and the percentage of dissatisfaction dropped by 70% and 60%, respectively, after the cooling system was installed. Ye et al. [12] also explored the influence of thermal comfort and worker productivity in factories, revealing that productivity reaches its maximum when the thermal sensation vote (TSV) of the subject is slightly cool instead of neutral or warm.

### 2.2. Robot Motion-Based Factors

In recent years, tremendous research efforts have been spent on human comfort evaluation and adaptation in HRC manufacturing tasks. The experiment task designs are usually based on robot motion-based factors, which include robot moving speed, the final position of object delivery, human–robot proximity, interaction time cost, robot movement trajectory, etc. Different individuals can have huge differences in their preferences. For example, some people prefer close-proximity interaction; others might prefer farther distances. Weitian et al. [30] proposed a computational Human Comfort Model (HuCoM) approach to model and quantify human comfort under HRC scenarios. To verify the proposed HuCoM model, Weitian et al. designed a series of model car assembly tasks based on four robot motion-based factors: robot speed, the position of object delivery, human–robot proximity, and left/right robot arm. The four primitive independent factors were adjusted to evaluate their influences on human comfort. Ross et al. [31] found that human comfort has a direct and immediate influence on the collaboration quality between the robot and its human partner and is also a significant factor for the robot to be aware of. Jessi et al. [32] developed a method of evaluating how the invasion of personal space by a robot affects human comfort. Przemyslaw et al. [33] examined human response to motion-level robot adaptation to determine its effect on team fluency, human satisfaction, and perceived safety and comfort. All research above proves that robot motion-related factors have critical impacts on human comfort during HRC tasks.

### 2.3. Anthropomorphism

Another factor is anthropomorphism, which refers to the attribution of a human being’s characteristic to a non-human object like robot [34,35]. The most well-known concept and the most important rule for robot appearance design is the “uncanny valley”, identified by the robotics professor Masahiro Mori in 1970 [36]. The discovery of the uncanny valley brought a huge change to previous understandings of the relationship between human emotional response and human likeness. Uncanny valley theory claims that the human emotional response only keeps increasing until it reaches a point beyond which the response quickly becomes strong revulsion. However, as the robot’s appearance continues to become even closer to a real human, the emotional response becomes positive once again. Such fall and rise changes form a valley-shaped curve in the relationship plot, as shown in Figure 1. Thus, the name “uncanny valley” is given.

For robotics appearance design, it is crucial to avoid uncanny valley in order to prevent giving people creepy feelings. Furthermore, simply avoiding falling into the uncanny valley range is not enough. Goetz et al. [37] proposed the hypothesis that people’s acceptance and cooperation with the robot can be improved by providing a better match between a robot’s social cues and its tasks. Minato et al. [35] supported Goetz et al.’s opinion by extending the original uncanny valley concept to a broader dimension which includes not only robot appearance but also behaviors. According to Minato et al.’s theory, the general evaluation of interaction benefits from good matchings of robot appearances and their corresponding behaviors. Bartneck et al. [34] also claimed that it is important to match the appearance of the robot with its abilities. A highly human-look-like robot might give the user the illusion that it is able to complete extremely complex tasks such as listening and talking, which it is not capable of.Therefore, robot developers need to be very careful in choosing the appearance design of their robots. MacDorman pointed out that appearance is not the only factor being able to trigger the uncanny valley effect [38]. Expectation violation and cognitive paradoxes can induce similar emotional reactions [39,40].

Despite the simplicity in understanding the concept of the uncanny valley, a great number of challenges in robot appearance design remain. There is a lack of a comfort model to predict which region of the uncanny valley curve the robot falls in. Thus, the difficulty and cost of robot appearance evaluation increase.

### 2.4. Robot Sociability

The last factor is robot sociability, which has been getting more attention recently. Applying social robots as mental health interventions for children has become increasingly popular in healthcare environments [41]. Kabacinska et al. found that robot interventions had positive impacts on children’s mental reactions; reduced depression and anger were found in testing children.

As social robots obtain increasing attention in the market and research field, scientists and engineers have started to look into more detailed sub-factors under robot sociability, such as their levels of animacy, likeability, perceived intelligence, and perceived safety; they have also studied how these sub-factors impact human comfort and reactions. Walters et al. [42] found that subject’s personality profiles influence personal spatial zones in human–robot interactions. People systematically prefer robots for jobs when the robot’s human likeness matches the sociability required in those jobs [34,35,37]. For example, a robot with a good manner and friendly speaking tone is preferred. Gasteiger et al. [43] listed four key factors in optimizing human experience during HRC tasks with social robots: (1) communication and language, (2) behavior and service, (3) proxemics, and (4) interface design.

Challenges in robot sociability factors still remain. Few studies quantitatively investigated why certain appearances of the robots are preferred. The relationship between appearance and comfort requires further research.

### 2.5. Discussion

In this section, the human comfort factors under HRC scenarios were reviewed in four categories: ergonomic factors, robot motion-based factors, anthropomorphism, and robot sociability factors. All cited works in this section are listed in Table 1. Papers are grouped and ordered based on influential factors. Short summaries of methodologies of each paper are also provided.

For ergonomic factors, the noise and thermal impacts on human comfort and how comfort varied along with these factors’ changes were introduced. These two factors are environmental factors, which are independent of the setup of robots. Methods of establishing noise and temperature models and evaluating annoyance levels were also presented. Thermal equations and on-body sensors were widely used to evaluate the body heat transfer data. However, limitations still exist in these methods. For the noise factor, not too many studies focus on the quantitative side, and most of the research work is based on subjective ratings only. Thus, a quantitative ergonomic factor-based comfort model would be greatly helpful for future research in this field. For the thermal factor, traditional methods use on-body sensors and thermal mannequins to collect data; however, the fact that different individuals have different rates of metabolism could create extra difficulties for thermal modeling and measuring. Another challenge of using mannequins is their low measuring accuracy, which does not correctly reflect human comfort levels.

The “uncanny valley” effect and corresponding theory have been found and widely used in analyzing comfort. Researchers have further expanded the original two-dimensional uncanny valley curve into three dimensions, claiming that human comfort will be further improved by matching the robot appearances and their corresponding behaviors. For robot-motion-based factors, a considerable amount of research has been done evaluating single-factor impact, including moving speed, the final position of object delivery, human–robot proximity, interaction time cost, robot movement trajectory, etc. These influences are highly dependent on individual preferences. Despite the fact that abundant research efforts have been made in this area, there is still a lack of a comprehensive comfort model which can handle multiple factors’ impact at the same time. It is very unlikely that human–robot interaction scenarios take place in reality that has only one varying factor. For robot sociability factors, an important rule is to avoid uncanny valley in appearance design. Human–robot proximity also plays a key role in human comfort in human–robot interaction. Similar to daily social interactions, a preferred interaction distance also exists between humans and robots. In terms of communication, people tend to prefer robots with better social skills and friendlier communications. Robots with better comprehension and communication skills, both physically and vocally, can greatly improve human comfort. The limitation in this area is that few studies have quantitatively investigated why certain appearance features and details of the robots are preferred. The relationship between appearance and comfort requires deeper investigations.

## 3. Review on Measurements of Comfort

The sense of comfort is a human’s subjective nature and results from a human’s reaction to the environment [45,46,47,48]. For human comfort measurement, there are two main widely used approaches: the self-evaluation approach (subjective measurements) and the physiological approach (objective measurements) [9].

### 3.1. Subjective Measurements

#### 3.1.1. Likert Scale

Questionnaires have been the most widely used data collection method in subjective rating measurements. Various kinds of rating scales have been used to assess a person’s subjective attitudes. Among all kinds of approaches, the Likert Scale is the most commonly used one to obtain scaled responses to a certain statement in survey research.Likert Scale was first introduced by psychologist Rensis Likert in his paper [49] in 1932, and thus the name Likert Scale was given. As Burns et al. stated in their book [50], when responding to a Likert Scale, respondents provide their level of agreement on a symmetric scale on a series of items. The Likert Scale will then capture the intensity of the subject’s feelings. Despite the advantages that questionnaires possess, they have limits as well. Questionnaires are typically not compatible with real-time data collection. Thus, in order to counteract this limitation, some researchers created hand-held devices.

#### 3.1.2. Hand-Held Device

Koay et al. developed a hand-held device equipped with a pressing button and pressure sensor as a measurement approach of subjective comfort levels in human–robot interaction experiments [51]. The test subject is instructed to press the button with different pressure and duration time to report his/her subjective feelings whenever discomfort feeling is perceived. In order to precisely match the button pressing moments and corresponding experiment events, time-stamped recording is required. Such a synchronization technique would be a great help in achieving more accurate analysis based on real-time data. Furthermore, besides the real-time data collection characteristics, the hand-held device also adds a new dimension to comfort data, which is discomfort duration. The duration length of the discomfort feeling can be used as a new feature for better comfort analysis. Wang et al. [52] designed another type of handheld subjective comfort collection device based on a single-chip microcomputer equipped with four buttons mapping to four comfort states. The device is used to collect real-time comfort data from passengers in a car ride.

Although the hand-held device has been proven to be useful in many cases, it still has many limitations and drawbacks. First, the disadvantage is the error input caused by the device’s sensitivity flaw; test subjects were found to accidentally press the button without notice [53]. To avoid this issue, subjects need to put their index fingers away from the button; however, this might introduce another problem where subjects might press too hard when suddenly encountering uncomfortable conditions. The second disadvantage of using the device is that the subjective pressing force inputs are difficult to maintain consistently and accurately to reflect corresponding comfort levels throughout a long-duration experiment. The third issue with using the device is that many subjects tend to forget to press the button after the experiment starts for a period of time while they are too focused on the tasks.

#### 3.1.3. Video Footage Analysis

Another widely used human comfort evaluation approach in HRI scenarios is the analysis of video footage which records the interactions between humans and the robots.

By viewing the videotapes, the event-related behaviors and activities of the subjects are counted and finally used in statistical analysis. For example, Salter et al. [54] used recorded video footage to analyze various types of children’s play styles with autonomous mobile robots by counting their body movement behaviors. Koey et al. [51] implemented a video annotation tool to mark and categorize specific human behaviors. Koey et al. used the time stamps information to sync the comfort data series, then matched the test subjects’ uncomfortable states shown in their video footage to determine the cause of discomfort in terms of robot behaviors.

Dautenhahn et al. [55] studied human micro-behaviors during human–robot interactions by recording the body reactions such as eye gazing and eye contact activities from children with autism. The videotaping method is even more helpful for scenarios where verbal communication and feedback are not applicable. The disadvantages of the video analysis technique are also not negligible. Firstly, video analysis is a highly-technique required skill that puts a strict standard on the person who carries out the task. Secondly, it is time-consuming and thus fatigue-inducing, which eventually might cause the video observer to overlook some critical details such as relevant behaviors and subjects’ facial expressions. Furthermore, even if the video observer did as best as he/she could, the facial or body expressions of the subjects might not fully or truly reveal his/her actual emotions.

### 3.2. Objective Measurements

Human bodies tend to present a variety of physiological responses such as respiration rate and blood pressure increase [56], skin temperature drop [57], heart rate variability (HRV), and pupil diameter quantitative characteristic changes [58,59,60]. Therefore, objective measurement methods mostly focus on these physiological signals.

#### 3.2.1. Heart Rate Variability

Among all the influential factors of human comfort, stress is one of the most important ones. Human stress level is closely related to heart rate-related indexes. When it comes to the quantitative study of heart rate, the first and the most critical concept to consider is heart rate variability, also known as HRV.

Heart rate is defined as the number of heartbeats per minute, while heart rate variability (HRV) represents the fluctuation in the time intervals between adjacent heartbeats [44]. HRV is a powerful tool in studying and monitoring psychological status changes due to the fact that HRV reflects the regulation of autonomic balance, blood pressure (BP), gas exchange, and possibly even facial muscles. Hilgarter et al. [61] found that HRV indices possess high sensitivities to psychological status fluctuations in stress response, regardless of age and sex.

The most commonly used methods of interpreting and processing HRV raw data are still categorized into two main groups—time-domain methods and frequency domain methods, although different types of other methods have been proposed over the years, such as geometric methods and nonlinear methods.

Time domain HR is an intuitive measurement of objective metrics. Several widely used time domain indexes are listed below [44,62]:Mean of Heart Rates;Standard Deviation of HRs;SDNN—the standard deviation of NN (normal-to-normal) intervals. It is often obtained over a 24-hour period since it is normally more accurate when measured over 24 h than short-period monitoring;SDANN—the standard deviation of the average NN intervals calculated over short periods;RMSSD—root mean square of successive differences, the square root of the mean of the squares of the successive differences between adjacent NNs;SDSD—standard deviation of successive differences, the standard deviation of the successive differences between adjacent NNs;NN50—the number of pairs of successive NNs that differ by more than 50 ms;pNN50—the proportion of NN50 divided by the total number of NNs.

Besides the methods of measuring HRV introduced above, other refined calculation methods have also been developed. Another simple measuring approach of HRV is the standard deviation of the mean R–R interval (SDRR) [63]. De Geus et al. [64] found that HR increases and SDRR decreases transiently when healthy subjects are acutely stressed. Berntson et al. [63] found that respiration also has a great impact on HR changes. Respiratory sinus arrhythmia (RSA) is considered an index of cardiac parasympathetic activity and tends to decrease under acute psychological stress [65,66]. Despite all the advantages and power that time-domain analyses possess, they are still limited in some cases, which results in requirements for other analysis approaches.

Frequency domain methods categorize heart rate oscillations into four bands as ultra-low-frequency (ULF), very-low-frequency (VLF), low-frequency (LF), and high-frequency (HF) bands, and then count the number of NN intervals that falls into each band. Based on the distribution of absolute or relative power, the Task Force of the European Society of Cardiology and the North American Society of Pacing and Electrophysiology [65] published a standard for this categorization [44].

Among these four bands, researchers tend to be more interested in LF and HF bands since their ratio provides a great amount of useful information. The ratio between LF power (0.05–0.15 Hz) and HF band power (0.15–0.4 Hz) reflects the instantaneous balance between sympathetic and parasympathetic activities [67]. A list of HRV-related index responses regarding mental stress is given below:Mean HR increases during mental stress [67];Mean RR-interval and RMSSD decrease as mental stress increases [68];LF/HF and LF tend to increase as mental stress increases [67,68];HF decreases during acute stress [69];LF/HF ratio is also important in evaluating thermal comfort. LF/HF ratio increases as the temperature get too hot or too cold;Heart rate variability decreases during stress.

Schubert et al. [67] designed a challenging speech task to induce acute psychological stress in order to study how the measures of heart rate and HRV can be affected by short-term stressors/long-term stress exposure. Schubert et al. found that SDRR, LF, and HF increased under acute stress, while RSA and LF/HF ratios remained still, and respiration rates decreased. The analysis of heart rate (HR) data for stress measurement is well known in physiological indexes [70,71]. Sawabe et al. [70] collected raw HR data with a thoracic HR band and an electrocardiograph circuit, and then obtained LF/HF ratio from the raw data. Eventually, comparisons on the LF/HF rate change within a few seconds were carried out to evaluate the stress level of a passenger during an autonomous vehicle simulation ride test. Wang et al. [72] used personal thermal sensation as a continuous function of time, and then adopted not only the time-domain and FFT features, but also applied the Hilbert Transform (HT) to extract the instantaneous amplitude (iA) of the LF and HF for thermal comfort modeling.

#### 3.2.2. Electrodermal Activity

The next physiological index is electrodermal activity (EDA), also known as skin conductance, galvanic skin response (GSR), electrodermal response (EDR), and skin conductance response (SCR). EDA is the property of the human body that causes continuous variation in the electrical characteristics of the skin. Skin resistance varies with the state of the sweat glands in the skin. The arousal of the sympathetic autonomic nervous system activity can result in the increase of the sweat gland, which leads to greater skin conductance. Thus, the EDA signal is widely used as another important index in evaluating a person’s psychological or physiological arousal in response to an external stimulus [73]. The higher the arousal, the higher the skin conductance. The change in skin response is linked with emotion, stress, and pain. As of today, EDA is considered the most popular method for studying human psychophysiological phenomena [74].

Skin conductance measurement is typically composed of two components—tonic skin conductance level (SCL) and phasic skin conductance response (SCR) that result from sympathetic neuronal activity. Tonic skin conductance levels can be considered as the smoothly and slowly changing levels, while the phasic skin conductance responses can be thought of as the rapidly changing peaks.

The EDA signals are usually collected at the palmar area of a subject’s hands or feet since these areas typically have the strongest sweat gland activities [73]. Skin conductance is captured using skin electrodes which are easy to apply. Data are acquired with sampling rates between 1–10 Hz and are measured in units of micro-Siemens (μS). Typical computing features of GSR include:Amplitude of SCR;Latency between stimulus and SCR onset;Recovery time of 63% amplitude;The distributions of the EDA peak height and the instantaneous peak rate.

Since the relationship between these GSR features and human comfort is very complex, there is no simple math formula to represent it. Thus, a great amount of research uses machine learning techniques to handle the problem. Shi et al. and Lagomarsino et al. [75,76] investigated the feasibility of using GSR to evaluate subjects’ cognitive loads. The GSR data results and analysis from the user experiments demonstrated that mean GSR across users increases as cognitive load increases. Khamaisi et al. [77] presented a strategy to evaluate the mental and physical workloads and stress of workers in heavy workload scenarios by measuring the EDA, HR, and eye activity signals. The experiment was set up in the VR environment. Villarejo et al. [78] used the GSR device and predicted whether the test subjects were in a mentally stressed situation with a success rate of 90.97%. Sawabe et al. [70] measured subjects’ personal skin conductance using the terminals of the subject’s two fingers. The eSense Skin GSR sensor was applied in their research to collect and analyze the EDA data. The stress response of the subject is detected by a rapid change in the GSR rate. Setz et al. [73] analyzed the effectiveness of using electrodermal activity (EDA) to distinguish stress from cognitive load under two designed stress factors. Multiple features were used in this research: (1) Mean, maximum, and minimum EDA levels; (2) Slope of the EDA level; (3) Mean EDA peak height; (4) Mean EDA peak rate in peaks/min; (5) Quantile thresholds at 25%, 50%, 75%, 85%, and 95% for the EDA peak height and the instantaneous peak rate. Then, the following classification methods were implemented: (1) linear discriminant analysis (LDA); (2) support vector machine (SVM); (3) nearest class center (NCC) algorithm. Setz et al. eventually found that EDA results successfully discriminate cognitive load from stress with an accuracy greater than 80%. Furthermore, the EDA peak height and the instantaneous peak rate were found to carry information about the stress level of a person.

#### 3.2.3. Skin Temperature

Skin temperature (SKT) measures the thermal changes on the skin. The fluctuations in skin temperature are mainly influenced by blood flow volume changes due to vascular resistance or arterial blood pressure variations. Local vascular resistance is mediated by the sympathetic nervous system [79]. Therefore, the SKT variation is another indicator of a person’s emotional state. Kim et al. [80] adopted a wide variety of physiological features for emotion classification, including maximum and mean skin temperatures within 50 s intervals. Zhai et al. [81] found that the patterns of temperature slope provide more meaningful information than the mean value in terms of emotion classification accuracy. Pao et al. [82] proposed a thermal sensation prediction model by adopting physiological features including body temperature, EDA, EEG, and ECG. The accuracy results indicate that this new model performs better than the predicted mean vote (PMV) model.

#### 3.2.4. Electroencephalography (EEG)

Electroencephalography, also known as EEG, is an electrophysiological process of recording the electrical activity of the brain by placing EEG electrodes on the surface of the user’s scalp. The electrical activity mainly comes from voltage changes from ionic current within and between some brain neurons. The collected signals will then go through processes such as amplifying, digitizing, and then being sent to a computer or mobile device for data processing [83].

The brain waves are usually divided into four bands by frequency: Delta, Alpha, Beta, and Gamma. The Delta band has the lowest frequency, while the Gamma band has the highest frequency [84]. Each band has its own features and carries specific information which reflects certain nervous system activity. For frequency band analysis and classification, power spectral analysis is implemented to visualize the EEG power of each frequency band. The differences in brain mapping of the relatively high-beta wave in the temporal lobe can be useful when assessing participants’ stress.

EEG is well known for a significant and reliable bio-signal reflecting mental fatigue. Cognitive loading induces mental fatigue, and people have difficulty processing visual stimuli or making decisions when they suffer from mental fatigue. The EEG signals can also be used to detect many high-level human emotions such as happiness, surprise, fear, disgust, etc. Current research mostly applies machine learning models such as SVM and RNN to extract time-domain or frequency-domain features from raw EEG signals and then implement classification.

Choi et al. [84] examined how indoor environmental elements such as temperature, odor irritants, and sound will impact human stress levels by designing multiple climate chambers and carrying out EEG tests to generate occupants’ brain maps. The experiment results demonstrated that brain wave analysis in the temporal lobe could be highly useful when assessing participants’ stress. Yao et al. [85] investigated the impact of environmental temperature changes on EEG, eventually finding that the β band is dominant under extreme temperature conditions, while α band power is significantly larger than the other bands under the neutral temperature condition. Kang et al. [86] created a wellness platform to address the visual discomfort issues generated in the stereoscopic 3D (S3D) display scenarios. The authors firstly determined the features that can be used as the index for visual discomfort perception, then applied machine learning techniques (SVM) to build a BCI framework to eventually optimize the S3D content based on the viewer’s EEG response. Lin et al. [87] studied predicting and categorizing four different human emotion states (joy, anger, sadness, and pleasure) during music listening based on recorded EEG signals. The authors found that most of the identified EEG features were extracted from the electrodes near the frontal and parietal lobes. Eyam et al. [88] proposed an approach which utilizes EEG to detect human emotional states and then instantly adapts COBOT parameters to human emotional states. The approach kept human emotions within a desirable range and increased the humans’ confidence and trust in the robot. Peng et al. [89] evaluated high-speed railway passengers’ overall comfort by using questionnaires and EEG through a series of field tests. The experimental results indicate that passengers have different neural signatures under different comfort states in the frequency and spatial domains. The β band is more relevant to comfort compared to others.

#### 3.2.5. Pupillometry

The last metric introduced in this section is pupillometry. Pupillometry is a reliable tool for studying cognitive and emotional processes, as well as for determining an individual’s emotional state [90,91]. The pupil is the black hole, also known as the aperture of the iris, which is located in the center of the eye that allows light to strike the retina. It is a pigmented structure that contains two antagonistic muscle groups—the sphincter and the dilator muscles [60]. The sphincter is responsible for constricting the pupil while the dilator is functioning to dilate the pupil. A great amount of research has shown that the extent of pupil diameter (PD) dilation is related to the mental effort load of the subject during cognitive tasks or psychological stresses [92,93,94].

Compared to other stress indexes such as cardiovascular activity, EDA, and EEG, the most significant advantage of pupillometry is its unobtrusiveness. No physical contact is required between the pupillary data collection devices and the human body. Typically, pupil activities can be simply measured with one video camera or more professional devices like an eye tracker. For example, the Tobii TX300 eye-tracking system measures eye movement at 300 Hz and another system—iView—developed by SensoMotoric Instruments with a 50 Hz sampling rate [91].

Previous research has demonstrated their effectiveness for pupillometry studies [95]. Some other eye-movement metrics, such as saccade parameters, are also found to be influenced by psychological stresses [96,97]. Zhang et al. [98] studied the relationship between colors in subway station and visual comfort through pupilometry analysis. The research proved that the pupillary unrest index and saccade rate in the eye movement index were significantly negatively correlated characteristics with the user’s comfort, which can be served as the evaluation parameters of visual comfort. Pedrotti et al. [60] studied the impact of psychological stress on pupillary activities by proposing a new method that utilized wavelet transform and neural networks. The experiment was based on simulated driving tasks; pupil diameter, EDA signals, and self-reported assessments were recorded. The neural network classifier proposed by the authors yielded 79.2% prediction accuracy among the four test scenarios.

Changes in pupil diameter have also been proven to be optimal in measuring human emotion. For example, pupil diameter increases when the person feels pleasure or fear. Babiker et al. [91] designed multiple experiment tests with positive and negative sound stimuli and recorded the pupillary responses from 30 participants. The pupillary measurements indicated that pupil dilation sharply increased during the sound stimuli tests, and the pupil dilation phenomenon was found to be even stronger for the negative stimuli scenarios.

### 3.3. Discussion

This section reviewed two main comfort measuring approaches—subjective measurement and objective measurement. All cited works in this section are listed in Table 2. Papers are grouped and ordered based on evaluation metrics. Short summaries of methodologies of each paper are also provided.

The subjective measurement approaches include the Likert-Scale evaluation, hand-held device, and video footage analysis, while the objective measurement approaches include utilizing human physiological signals such as heart rate variability (HRV), electrodermal activities (EDA), skin temperature, electroencephalography (EEG), and pupillometry. Since comfort is widely accepted by most researchers as a subjective mental response to environmental stimulus [14], subjective rating approaches such as the Likert Scale are usually conceived as the most accurate type of human comfort measuring method so far. Thus, subjective ratings are commonly used as ground truth values in many comfort prediction models which utilize physiological signals from the human body as model inputs. Despite the advantages questionnaires and the Likert Scale possess, they have limits as well. Questionnaires are not capable of real-time data collection; in addition, the comfort data are highly discrete, which sacrifices accuracies in true comfort reflections. Fortunately, hand-held devices counteract the issue. However, it also creates new challenges such as the button sensitivity issue, accuracy issue, and subject’s focus problem. Particularly in HRC tasks, it is sometimes impossible for the subjects to press the button while executing the required actions. Therefore, a third method can be used as the compensating method. Video footage analysis has been proven to be effective in capturing real-time human reactions, which can be very useful in analyzing his/her emotion, but it is a highly-technique-required skill and fatigue-inducing job.

HRV and EDA signals have been widely used to assess long-term and short-term psychological states, which can take from several minutes to 24 h. Time-domain features of HRV analysis and EDA analysis can be easily implemented in HRI scenarios; however, frequency-domain methods of HRV analysis sometimes can be tricky to apply. The reason is that these frequency-domain features usually require at least up to five-minute recording to be effective, but many HRI tasks only last a short period of time. This greatly reduces the chance of frequency-domain features being used. EEG features have also shown advantages in predicting human emotion and cognitive load. Despite the fact that a great amount of research has proved the effectiveness of utilizing bio-signals for stress, comfort, and emotion measurements, the physiological signals are susceptible to noises and uncertainties, which could be affected by many unknown factors and random events. In addition, there is still a lack of a complex model which directly relates body signals to more general and comprehensive human comfort ratings.

## 4. Comfort Improvement Approaches

The subjective nature of comfort leads to individual differences in preferences of robot behaviors. In general, adapting the robot’s performance to humans can result in a positive impact on one’s comfort. A great deal of previous research has focused on improving human comfort feedback during HRC tasks using motion-based, social factor-based approaches, and other typical methods.

### 4.1. Motion-Based Improvement Methods

Robot adaptability refers to the ability of a robot to adjust its working style and responses based on the environmental change and stimulus in order to better achieve the task goal. Robot adaptability consists of many factors such as robot pose, speed, moving trajectory adaptations, as well as adaptations with respect to social factors such as voice and gesture.

#### 4.1.1. Optimizing Robot Moving Trajectory

An optimized robot movement trajectory that clearly expresses the robot’s intent and matches the human partner’s expectation will lead to more fluent collaborations and higher human comfort. Dragan et al. [99] designed an HRC task that requires the human subject to collaborate with the robot for tea serving. Three types of robot moving trajectories were created, and the results showed that the most predictable type of motion obtained the highest user score ratings and the least time cost. Alami et al. [100] proposed a framework, which allows the robot to select and perform its tasks based on the human partner’s presence, needs, and preferences. The framework introduced two criteria, the security criteria and the visibility criteria, which prevent the robot from approaching too close to humans and also ensure the visibility of the selected path. Gielniak et al. [101] developed an autonomous algorithm that creates anticipatory motion variants from a single motion exemplar that has hand and body symbols as a part of its communicative intent. The results demonstrated that humans understood robot intent sooner than motions without anticipation. Dinh et al. [102] presented a framework that generates predictable robot motions with dynamic obstacle avoidance during human–robot interactions by using the policy improvement method. Besides using Dynamic Motion Primitives for trajectory generation, an additional potential field term was added to penalize trajectories which could lead to collisions. A cost function is designed to minimize the risk of collisions and maximize the predictability of robot motions.

Human awareness of COBOTs is not only critical due to safety concerns but also because of better human–robot collaboration experience and efficiency. Lasota et al. [33] applied a PhaseSpace motion capture system to keep track of the human arm’s position during a screw-tightening task while the system predicts the intent of the human subject and estimates the shared workspace, then adjusts the robot trajectory to avoid the collision. Both quantitative measurements and subjective feedback indicated that subjects preferred the human-aware setup over the baseline setup and had a higher perceived comfort level and higher working efficiency.

#### 4.1.2. Planning Robot with Adaptive Poses

Human workers in traditional factory working environments usually repeat certain postures and movements constantly, which could cause certain aggravated work-related diseases. Such diseases are typically known as “Musculoskeletal disorders” (MSDs), which are also the largest category of work-related diseases [103]. New adaptive robots in the next generation should be able to prevent these diseases and discomfort feelings from human workers.Ciccarelli et al. [104] proposed a system to improve human postural comfort by optimizing robot behavior. The system is based on workers’ anthropometric characteristics, posture monitoring, task requirements, and a real-time risk assessment by standard methodology. Busch et al. [105] investigated and developed the approach to improve the human worker’s collaborative posture during HRC tasks. The authors integrated the REBA method [106] into a framework that estimates the ergonomic costs of each human body joint. The cost function and postural assessment techniques are taken from the ergonomic research. For the cost calculation, each joint has an associated value which represents the MSD risk score. The final optimization objective is to minimize the overall risk scores of the human body. Eventually, optimal robotic behaviors which guide human workers to better postures were derived based on the framework.Tassi et al. [107] developed a novel Augmented Hierarchical Quadratic Programming (AHQP) framework which integrates human-related parameters to optimize ergonomics, for multi-tasking control in Human–Robot Collaboration. The framework combines typical industrial manufacturing parameters (e.g., cycle times, productivity) and human comfort (e.g., ergonomics, preference), in order to identify an optimal trade-off.

Despite the great work from Busch et al., Chen et al. [108] discovered that optimizing only muscular comfort is not sufficient. For example, while the human may have better muscular comfort, he or she can be dangerously close to the robot and is obstructed by the robot links. Thus, Chen et al. presented a planning algorithm for robot grasping and positioning to improve both human comfort and safety. The algorithm considers both the muscular activation level required to carry out the task and the human spatial perception during the interaction. By maximizing both comfort criteria, both the grasp stability and human comfort were improved.

### 4.2. Sociability-Based Improvement Methods

The constantly aging population structure and shortages in healthcare resources in many countries have greatly promoted the research and the application of nursing robots and social companion robots in the past decades. Previous research has shown that humans tend to accept a robot more easily with better social abilities and behaviors [109].

Social robot acceptance typically can be categorized into two branches—functional acceptance and social acceptance [110]. Functional acceptance mainly refers to the human’s acceptance level of the robot’s usability, while social acceptance refers to whether the human is willing to build a pet-like relationship with the robot or become a conversational partner.

Heerink et al. [110] investigated users’ preference and acceptance of robots with different levels of social abilities and behaviors. The five basic features (Cooperation, Empathy, Assertion, Self-Control, and Responsibility) from the Gresham and Elliott’s Social Abilities Rating System (SSRS) [111] were used as correlated features corresponding to certain behaviors programmed into the robot. The iCat robot was used as an interactive robot and had two working conditions. One of the conditions was more socially communicative with more facial expressions and head nodding, etc. Results demonstrated that participants generally had a higher preference for the more socially communicative setup of the robot and tended to be more willing to interact with it.

From the perspective of robot sociability, developing appearances for robots in human–robot interaction, especially for domestic service robots and health care robots in public settings, plays an important role in improving human comfort [42,110,112]. Walters et al. found that 60% of human subjects prefer robot-approaching distances that are expected for normal social interactions between humans [42]. As mentioned in Section 2, human–robot proximity also has great influence on human comfort. Jessi et al. [32] built a testbed based on a Baxter humanoid robot and Wizard of Oz implementation; they then evaluated how the invasion of personal space by a robot, with appropriate social context, affects human comfort.

Kuo et al. [113] studied the influence of age and gender factors on the acceptance of healthcare robots in HRI scenarios. The differences in ages between the two groups are barely noticeable, but a significant gender-driven difference was found. Van Dijk [114] found that letting the elders discover the convenience and usefulness of the devices would help increase elderly people’s acceptance. Mitzner et al. [115] also proposed similar findings about the benefits and rewards of letting elderly people have a positive experience with the technologies.

### 4.3. Other Typical Methods

Wang et al. [116] proposed a Teaching-learning-prediction model to let the robot learn from human demonstration. Robot action selections are based on human intention anticipation. Shah et al. [117] made the robot emulate the effective coordination behaviors observed in human teams to minimize the human’s idle time. Hoffman et al. [118] proposed the concept of a perceptual symbol system, which uses simulation and inter-modal reinforcement to allow for decreased robot reaction time. Robot emulates the Perceptual-symbol practice in robot decision-makings.

### 4.4. Discussion

This section reviewed several types of comfort improvement methods, including the motion-based method, sociability-based method, and some other typical methods. All cited works in this section are listed in Table 3. Papers are grouped and ordered based on evaluation metrics. Short summaries of methodologies of each paper are also provided.

The motion-based methods include trajectory optimization and pose adaptation approaches. Trajectory optimization algorithms mainly focus on two ways to improve human comfort—actively adapting robot trajectories to human arm motions to provide humans higher trust and thus higher comfort; adapting trajectories to better match human expectations to improve collaboration fluency and thus provide higher comfort response. Pose adaptation methods aim at adapting robot delivery poses to reduce musculoskeletal disorders-related diseases to improve human joint comfort.

Previous research pointed out that humans tend to have a preference on a robot with better social abilities and behaviors. Thus, the methods of improving social robot acceptance are typically focused on two branches—functional acceptance improvement and social acceptance improvement. Functional acceptance improvement methods mainly focus on improving the usability of the robots, while the social acceptance improvement methods focus on the effectiveness of improving robots’ communication skills and appearances, as well as adapting the interactive distance with humans. In social robot designs, age and gender factors need to be taken into consideration. Some researchers found that a huge gender-driven difference exists on the acceptance of healthcare robots. In order to improve the elder people’s acceptance, the most effective way is to let them realize the convenience and usefulness of these machines and devices. This chapter also covers some typical methods which will enhance robot collaboration efficiency to improve the human experience during the tasks. For example, a teaching–learning–prediction model enables the robot to learn from human demonstration. A robot can minimize the human’s idle time by observing and studying from human’s effective behaviors.

Most methods introduced above are empirical, and there is still a lack of theoretical comfort model-guided comfort improvement methods. In addition, the collaborative task designs in existing studies are usually composed of only one or two simple moves from the human side, which can not accurately simulate some of the real-world manufacturing scenarios. For the test of robot appearance designs, adopting virtual reality technology seems to be a better approach with higher freedom of customization and lower cost. In addition, most methods above only study improving comfort based on some specific and limiting factors. Controlling multiple different factors to improve the general comfort level remains a challenge. In general, the amount of research on the topic of human comfort improvement is much less than the two previous topics: influential factors and measurement methods. However, the findings and achievements in these two topics enable us to better understand the human comfort and also build a foundation for future research work in improving human comfort in HRC. For future research works, more efforts should be focused on comfort improvement methods which consider multiple factors simultaneously, as well as methods that merge real-time subjective comfort measurement and physiological signal-based measurement methods to improve comfort in real time during HRC tasks.

## 5. Conclusions

Three major research topics on human comfort in human–robot collaboration scenarios were reviewed in this paper. In Section 1, the background of current manufacturing environment setups was introduced, and the usage of collaborative robots still remains a small portion. One of the main concerns preventing COBOTs from becoming a big part of the industry is the relatively low user acceptance. In order to improve the user perceived comfort, safety, and trust in COBOTs, a great amount of research has been done during the past few decades. The influential factors on human comfort during HRC tasks were introduced in Section 2, including ergonomic factors, motion-based factors, anthropomorphism, and robot sociability factors. In Section 3, human comfort measurement methods which consist of subjective and objective measurement approaches were reviewed. Section 4 covers the comfort improvement methods, including robot motion-based approaches, sociability-based approaches, and other typical methods.

The human comfort factors under HRC scenarios can be classified in four categories: ergonomic factors, robot motion-based factors, anthropomorphism, and robot sociability factors. Ergonomic factors are independent of the setup of robots. Robot-motion-based factors, including moving speed, the final position of object delivery, and human–robot proximity, are highly dependent on individual preferences. Robot sociability factors also play a key role in human comfort. Robots with better communication skills can greatly improve human comfort. Comfort measurement methods consist of two main branches—subjective measurement and objective measurement. The subjective measurement approaches include the Likert-Scale evaluation, hand-held device, and video footage analysis, while the objective measurement approaches include utilizing human physiological signals such as heart rate variability (HRV), electrodermal activities (EDA), skin temperature, electroencephalography (EEG), and pupillometry. The subjective measurement results are usually considered as the ground truth values and considered to be more reliable than physiological measurement results. Likert Scale is the most accurate approach but lack of real-time data acquisition ability, while hand-held device and video footage analysis methods provide real-time data but sacrifice reliability and accuracy. Physiological signals have been widely used to assess long-term and short-term psychological states, emotions and cognitive loads, which can take from several minutes to 24 h. Features extracted from these physiological signals usually consist of two types—time-domain features and frequency-domain features. Comfort improvement methods, including the motion-based method, sociability-based method, and some other typical methods, can improve human comfort by adapting robots’ trajectories, poses, communication styles, and even appearances.

There are also some promising future research directions based on this review. Firstly, more unknown influential factors can be explored in the context of HRC. Secondly, better objective measurement approaches are needed. Most of the physiological metrics for objective comfort measurement introduced in this paper only demonstrate the relationship between body signals and specific comfort factors. A complete model needs to be developed to better map these body signals to general and comprehensive comfort ratings. Thirdly, for comfort improvement methods, there is still a lack of theoretical comfort-model-guided methods. Most methods only study improving comfort based on some specific and limiting factors. Controlling multiple different factors to improve the general comfort level remains a challenge. Therefore, more research work is expected to understand, measure, and improve human comfort in the context of human–robot collaboration.

## Figures and Tables

**Figure 1 sensors-22-07431-f001:**
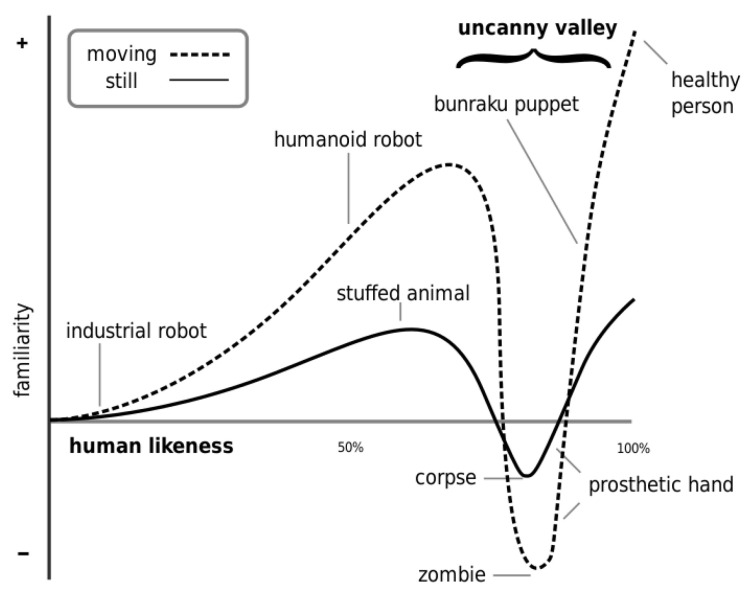
Uncanny valley plot [36].

**Table 1 sensors-22-07431-t001:** Influential factors of human comfort.

Author / References #	Factors	Methodologies
Guarnaccia et al. (2014) [18,19]	Noise	Noise Source Characterization; Noise Level Measurements
Ouis (2001) [20]	Noise	Noise Source Characterization; Sound Pressure Level Measurement;
		Acoustical characteristics of traffic noise;
		Measurement of Annoyance and Discomfort from Noises
Hall et al. (1985) [21]	Noise	Proposed a model demonstrating how activity interference affects the probability of annoyance
Izumi and Yano (1991) [22]	Noise	Developed a ‘path analysis’ to explain the annoyance responses obtained from questionnaires
Pennig et al. (2012) [23]	Noise	Subjective Measurement; Questionnaire
ASHRAE Standard 55 [24]	Thermal	Definition of Thermal Comfort
ASHRAE Handbook of Fundamentals [25]	Thermal	Energy Balance Equation for the Human Body
Da Silva (2002) [26]	Thermal	Thermal mannequins; heat conduction mathematical model; Sound Chamber; Combination of Subjective & Objective Measurement
Ormuž et al. (2004) [27]	Thermal; Noise	Thermal mannequins; Sound Testing Chamber;
		Combination of Subjective & Objective Measurement
Tiller et al. (2010) [18]	Thermal; Noise	Subjective Measurement (Likert Scale Rating); Questionnaire
Weitian et al. (2018) [30]	Motion-based	Proposed a computational model to quantify the human comfort;
		Subjective Measurement;
Ross et al. (2015) [31]	Motion-based	Human-Robotic Interaction Tasks; Combination of Subjective & Objective Measurement
Jessi et al. (2018) [32]	Motion-based	Human-Robotic Interaction Tasks; Wizard of Oz;
		Combination of Subjective & Objective Measurement;
Lasota et al. (2014) [33]	Motion-based	HRC-Tasks Experiments;
		Adjust the robot movement trajectories and moving speed based on test subjects’ reactions
Bartneck et al. (2009) [34]	Anthropomorphism	HRC-Tasks Experiments; Subjective Measurement
Minato et al. (2005) [35]	Anthropomorphism	Human-Robotic Interaction Tasks; Combination of Subjective & Objective Measurement
Masahiro Mori (2012) [36]	Anthropomorphism	Thought Experiment
MacDorman (2006) [38]	Anthropomorphism	Interview; Questionnaire
Goetz et al. (2003) [37]	Robot Sociability	Human-robotic Communication Tasks; Objective Measurement; Questionnaire
Katarzyna et al. (2020) [41]	Robot Sociability	Review on examining the impacts that social robots such as Nao, Paro, Huggable, Tega imposed on children in various scenarios.
Gasteiger et al. (2021) [43]	Robot Sociability	A review of key factors influencing human experience in HRC
Walters et al. (2005) [44]	Human–Robot Proximity; Sociability	Human-Robotic Interaction Tasks; Combination of Subjective & Objective Measurement

**Table 2 sensors-22-07431-t002:** Measurement methods and metrics of human comfort.

Author / References #	Metrics	Methodologies
Hart et al. (1988) [48]	Task Load	Likert Scale based Questionnaires
Haspiel et al. (2018) [45]	Trust; Anxiety; Preference; Cognitive Load	Autonomous Vehicle Ride Simulation; Likert Scale based Questionnaires
Peterson et al. (2017) [46]	Situational Awareness; Trust	Autonomous Vehicle Ride Simulation with Secondary Task; Questionnaires, Eye-tracking, Heart Rate, Galvanic Skin Response
Peterson et al. (2018) [47]	Perceived Risk; Trust	Autonomous Vehicle Ride Simulation with Secondary Task; Questionnaires, Eye-tracking, Heart Rate, Galvanic Skin Response
Koay et al. (2005) [51]	Self-reported Value	Human-Robotic Interaction Tasks; Hand-held Device, Questionnaires
Wang et al. (2020) [52]	Self-reported Value	Ride Comfort; Hand-held Device, Questionnaires
Su et al. (2021) [53]	Hand-held Device; Self-reported Value; EDA; EEG; Pupilometry	Ride Comfort; Subjective & Objective Measurements
Salter et al. (2004) [54]	Behavior Preference	Human-Robotic Interaction Tasks; Recorded video footage
Dautenhahn et al. (2002) [55]	Micro-behaviors	Recording body reactions during human–robotic interaction tasks
Wei (2013) [56]	Stress	Respiration (RSP); Electromyogram (EMG)
Ramos et al. (2014) [59]	Stress	Heart Rate (HR); Respiration Rate; skin temperature; EDA
De Geus et al. [64]	Stress	Impact of Stress on Heart rate variability (HRV) Metrics
Setz et al. (2009) [73]	Cognitive Load; Stress	Memory Tasks for Human; Galvanic Skin Response, Linear Discriminant Analysis, SVM;
Shi et al. (2007) [75]	Cognitive Load; Stress	Cognitive Load and Stress Inducing Tasks; Electrodermal Activity
Lagomarsino et al. (2022) [76]	Cognitive Load	Cognitive Load; HRC Tasks; Electrodermal Activity
Kaklauskas et al. (2011) [57]	Emotion; Work Productivity	Heart Rate; Blood Pressure; Skin Temperature; Skin Conductance
Zhang et al. (2014) [58]	Cognitive Workload	EEG; EDA; Heart rate variability (HRV); Cognitive Load Experiment
Shaffer et al. (2017) [44]	Heart Rate Variability	Heart rate variability (HRV) Metrics and Features
Hilgarter el al. (2021) [61]	Heart Rate	Verbal Learning Task; Questionnaires
Berntson et al. (1997) [63]	Heart Rate Variability	Heart rate variability (HRV) Metrics and Features
Task Force of the European Society of Cardiology and the North American Society of Pacing and Electrophysiology [65]	Heart Rate Oscillation	Standard for Categorization of Heart Rate Oscillation Bands
Schubert et al. (2008) [67]	Heart Rate Variability	Chronic and Short-term Stress Effects on Heart Rate Variability (HRV)
Castaldo et al. (2015) [68]	Heart Rate Variability	Acute mental stress and short term Heart Rate Variability (HRV) measures in time, frequency and nonlinear domain
Pagani et al. (1997) [69]	Heart Rate Variability	Relationship between HRV Components and Nerve Activity
Sawabe et al. (2018) [70]	Stress; Heart Rate; Galvanic Skin Response	Autonomous Vehicle Ride Simulation; Heart Rate; Galvanic Skin Response;
Wang et al. (2022) [72]	Heart Rate Variability	Thermal Comfort Experiments; FFT, time-domain, HT features
Boucsein (2012) [74]	EDA	Physiological States
Villarejo et al. (2012) [78]	EDA; Stress	Emotion Inducing Tasks; Math and Reading Tasks; EDA;
Jang et al. (2015) [79]	EDA; Emotions of boredom, pain, and surprise	Emotion Stimulation Tasks; ECG; EDA; Skin Temperature;
Khamaisi et al. (2022) [77]	EDA; Stress; HRV; Pupilometry	VR Simulation; Worker Mental Stress under Heavy Workload
Kim et al. (2004) [80]	Skin Temperature; EDA; Emotions Detection; HRV; Pupilometry	Multimodal (audio, visual and cognitive) approach to evoke specific emotional status
Zhai et al. (2006) [81]	Skin Temperature; EDA; Stress; Pupil Diameter	Stress induction interactive tasks; SVM;
Pao et al. (2022) [82]	Skin Temperature; Thermal Comfort	Skin Temperature; EDA; EEG; ECG; Thermal Chamber Experiment
Choi et al. (2015) [84]	EEG; Stress	Human in a Stress Test Chamber; Paper-based Test; EEG-based Test;
Yao et al. (2008) [85]	EEG; Thermal Comfort	Climate Chamber; Questionnaires; Skin Temperature; EEG; ECG
Lin et al. (2010) [87]	EEG; Emotion	Music Listening Tasks
Eyam et al. (2021) [88]	EEG; Human Emotional States	HRC Tasks; EEG
Peng et al. (2022) [89]	EEG; Passenger Overall Comfort	Field Tests; EEG
Kang et al. (2017) [86]	EEG; Visual Comfort	Stereoscopic 3D video; EEG Response; SVM;
Granholm et al. (2004) [90]	Pupillometry; Cognitive and Emotional Process	Cognition and Emotion Inducing Tasks;
Pedrotti et al. (2014) [60]	Pupillometry; Stress; EEG;	Simulated Driving Task; EEG Response; Questionnaire; Neural Network;
Babiker et al. (2015) [91]	Pupillometry; Emotion Detection;	Audio Stimulation; Pupil Response; Subjective Ratings; Machine Learning; kNN;
Beatty (1982) [92]	Pupillometry; Mental Effort Load	
Bradley et al. (2008) [94]	Pupillometry; Emotional Arousal	Picture-viewing Tasks; Pupil Diameter; EDA; Heart Rate;
Klingner et al. (2008) [95]	Pupillometry; Cognitive Load;	Task-evoked Pupillary Response; Remote Eye Tracker
Minin et al. (2011) [96]	Pupillometry; Stress; Eye-movement;	Simulated Driving Task (Lane Change); Visual Search Task;
Zhang et al. (2022) [98]	Pupilometry; Visual Comfort	Pupillary Unrest Index & Saccade Rate in the Eye Movement

**Table 3 sensors-22-07431-t003:** Comfort improvement methods.

Author / References #	Metrics	Methodologies
Dragan et al. (2015) [99]	Anticipatory Robot Movement Trajectory	HRC-Tasks Experiments; Combination of Subjective & Objective Measurement
Gielniak et al. (2011) [101]	Anticipatory Robot Movement Trajectory	HRC-Tasks Experiments; Combination of Subjective & Objective Measurement
Dinh et al. (2019) [102]	Anticipatory Robot Movement Trajectory	HRC-Tasks Experiments; Black-box Optimization, Dynamic Motion Primitives, Policy Improvement
Ciccarelli et al. (2022) [104]	Robot Poses Optimization	HRC-Tasks Experiments; Muscular comfort Optimization
Busch et al. (2017) [105]	Robot Poses Optimization	HRC-Tasks Experiments; Objective Measurement; Questionnaires; Muscular comfort Optimization;
Tassi et al. (2022) [107]	Robot Poses Optimization	HRC-Tasks; Trade-off between human comfort and Task Efficiency; Muscular comfort Optimization
Chen et al. (2018) [108]	Robot Poses and Position Optimization	HRC-Tasks Experiments; Objective Measurement; Muscular comfort and Human Spatial Perception Optimization;
Alami et al. (2005) [100]	Human-aware robot motion	High-level Symbolic Planning
Lasota et al. (2014) [33]	Human intention anticipation;	HRC-Tasks Experiments; Combination of Subjective & Objective Measurement
	Human-aware robot motion;	Adjust the robot movement trajectories and moving speed based on test subjects’ reactions
	Adaptive robot speeds	
Jessi et al. (2018) [32]	Adaptive Human–Robot Proximity	Human-Robotic Interaction Tasks; Wizard of Oz;
		Combination of Subjective & Objective Measurement;
Ruyter et al. (2005) [109]	Robot Sociability; Robot Communication skills	Home Dialogue System; Wizard of Oz experiment; Robotic interface simulating human social behaviors
Walters et al. (2005) [42]	Robot Sociability; Human–Robot Interactive Distance	Human–Robot Interaction Experiments; Combination of Subjective & Objective Measurement
Heerink et al. (2006) [110]	Robot Sociability; Robot Communication skills	Human–Robot Communication Experiments; Subjective Measurement—3-point scale Questionnaires
Kuo et al. (2009) [113]	Robot Sociability; User Acceptance	HRC-Tasks Experiments; Objective Measurement; Questionnaires
Wang et al. (2018) [116]	Human intention prediction	Teaching-learning prediction (TLP) model based on extreme learning machine (ELM) algorithms using online natural multi-modal information for the robot to learn from human hand-over demonstrations and predict human intentions
Hoffman et al. (2008) [118]	Human intention prediction; Robot decision-makings	A perceptual symbol system, which uses simulation and inter-modal reinforcement to allow for decreased reaction time through top-down biasing of perceptual processing.
Shah et al. (2011) [117]	Human-inspired robot task execution	A task-level executive that enables a robot to collaboratively execute a shared plan with a person. The system chooses and schedules the robot’s actions, adapts to the human partner, and acts to minimize the human’s idle time.

## Data Availability

Not applicable.

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
