# Peer review of "A Review on Human Comfort Factors, Measurements, and Improvements in Human–Robot Collaboration"

_sensors, 2022, doi:10.3390/s22197431_

Round 1

Reviewer 1 Report

In general, the contribution of this manuscript is very good. Therefore, my recommendation is to accept (major revision) this manuscript that has reference sensors-1920485 after several adjustments must be made before publication.

My specific comments is:

In tables 1, 2, and 4, please put two new columns that will demonstrate the limitations of each algorithm or the work and how to solve it.

Author Response

Response to Reviewer 1 Comments

In general, the contribution of this manuscript is very good. Therefore, my recommendation is to accept (major revision) this manuscript that has reference sensors-1920485 after several adjustments must be made before publication.

Major comments:

In tables 1, 2, and 4, please put two new columns that will demonstrate the limitations of each algorithm or the work and how to solve it.

Response:

Thank you for the valuable comment! Adding limitations and solutions is a great suggestion. Considering the consequence that the table will be too crowded and difficult to read by adding two columns and the fact that some papers within the same group of factors/metrics have very similar limitations and potential solutions, we have added the limitations and solutions in the discussion subsection after each table.

The revision can be found in Page 7 Paragraph 3-4, Page 15 Paragraph 4-5, and Page 19 Paragraph 5.

Reviewer 2 Report

The manuscript presents the review well but the latest references are missing. It's the end of 2022 and very few latest work has been cited especially from the last three years. I don't think this would be of any interest to the readers if the authors don't focus on research from the last five years.  

Author Response

Response to Reviewer 2 Comments

Major comments:

The manuscript presents the review well but the latest references are missing. It's the end of 2022 and very few latest work has been cited especially from the last three years. I don't think this would be of any interest to the readers if the authors don't focus on research from the last five years.  

Response:

Thank you for the valuable comment! We have expanded the review by adding the latest references in the past two years.

The revision can be found in Page 6 Paragraph 4, Page 8 Paragraph 3, Page 12 Paragraph 1, Page 12 Paragraph 5, Page 13 Paragraph 2, Page 14 Paragraph 2, Page 14 Paragraph 5 and Page 16 Paragraph 4.

Reviewer 3 Report

The paper is good - in many fields there is a lack of research on evaluation methods for various phenomena, so the paper is meeting this need to some degree. 

However, there is a lack of description of what comfort actually is. It is stated that is it subjective, and that it can be affected by various factors. But what does it constitute of? It is mentioned at the end discussion that factors such as understandability, acceptability, predictability, security etc. are important factors, and I would have liked to see a more holistic discussion of this already in the background. Without that solid start, one cannot go further and discuss how to measure it. 

Title is a bit fuzzy at the end - what kind of improvements? Some wordings are off, and sometimes the grammar. Tables need better readability, and shouldn't they be ordered somehow - either based on researchers or factor? The discussion at the end could be a bit improved as well, discussing the challenges and outlining future work. 

Author Response

Response to Reviewer 3 Comments

The paper is good - in many fields there is a lack of research on evaluation methods for various phenomena, so the paper is meeting this need to some degree. 

Major comments:

  1. There is a lack of description of what comfort actually is. It is stated that is it subjective, and that it can be affected by various factors. But what does it constitute of? It is mentioned at the end discussion that factors such as understandability, acceptability, predictability, security etc. are important factors, and I would have liked to see a more holistic discussion of this already in the background. Without that solid start, one cannot go further and discuss how to measure it. 

Response:

      Thank you for the valuable comment! A definition introduction of comfort has been given in Section I as “This review paper focuses on human comfort defined as a feeling of ease in human-robot collaboration contexts.” In addition, a discussion of different opinions to human comfort is presented. Despite of some disagreements from different researchers, consensus has been achieved as: (1) comfort is subjectively determined by each individual’s personal nature; (2) comfort can be affected by a wide variety of factors from multiple natures such as physical, physiological or psychological; and (3) comfort is affected by one’s reaction to the environment stimulus.

      The revision can be found in Page 2 Paragraph 4.

  1. Title is a bit fuzzy at the end - what kind of improvements?

Response:

      Thank you for the valuable comment! To make it more clear, we have changed the title to “A Review on Human Comfort Factors, Measurement, and Improvement Approaches in Human-Robot Collaboration”.

  1. Some wordings are off, and sometimes the grammar. Tables need better readability, and shouldn't they be ordered somehow - either based on researchers or factor?

Response:

      Thank you for the valuable comment! We have proofread the paper by correcting the misspellings, typos and grammar errors. We have also grouped the Table 1 by factors and Table 2 and 3 by metrics. 

      The revision can be found in Table 1, 2 and 4.

  1. The discussion at the end could be a bit improved as well, discussing the challenges and outlining future work. 

Response:

      Thank you for the valuable comment! We have added more discussion on the challenges and future work.

      The revision can be found in Page 19 Paragraph 5.

Minor comments:

  1. In the abstract you state "approaches", but still vary vague. Approaches in terms of what?

Response:

Thank you for the valuable comment! The comfort improvement approaches refer to ways of improving human comfort during Human-robot collaboration. We have introduced such approaches in Section 5 such as optimizing the robots’ trajectories and poses, adjusting the robots’ appearances and communication styles, and making robots act based on prediction of human intentions. To make it more clear, we have changed the description to “human comfort improvement approaches” in the abstract.

  1. Multiple Grammar and writing suggestions in the attached pdf file.

Response:

Thank you so much for your patient and detailed suggestions. We have fixed all of these writing issues and deleted the redundant formula, as well as added a new figure for the uncanny valley subsection. Horizontal lines have been added to table 1, 2 and 4 to improve the readability of the tables.

In the thermal factor subsection, the statement, “…the most preferred temperature range is within 72 - 76 °F, …” uses the Fahrenheit unit system. The reason we chose not to change it to the SI units is that we wanted to keep it consistent with the original paper’s content and unit system.

Round 2

Reviewer 1 Report

The contribution of this manuscript is excellent. The author’s answer was very good and satisfied. Therefore, my recommendation is to accept the manuscript that has No. sensors-1920485 for publication.

Reviewer 2 Report

Thank you for addressing the comments.